# A PATCH LEVEL PERSPECTIVE OF PROMPT TUNING

## ABSTRACT

Prompt tuning is an efficient way to adapt large foundation models. It introduces learnable prompts with the input data tokens, offering a practical alternative to full model finetuning. However, when prompts are trained on base/target tasks, they often overfit, leading to reduced performance on novel, unseen tasks. To address this limitation, various techniques leverage global image semantics to improve accuracy on unseen tasks while maintaining performance on base tasks. However, they often overlook the rich fine-grained local information that could be crucial for capturing finer semantics and improving generalisation. In this work, we propose a modular approach to prompt tuning that leverages local semantics by incorporating patch-level information, representing the first integration of such semantics in this context. Specifically, we integrate patch-level information across vision, text, and predictions through three consistency mechanisms: 1) Patch-based consistency loss that aligns patches from the prompted input image with those from the same image processed by a frozen model, while also enforcing inter-view consistency by applying the loss across different views, capturing fine-grained regional dependencies and improving vision representation quality; 2) Text prompt consistency loss, where view-specific text prompts are tailored and regularised to maintain coherence across views; 3) Vision features for each view, enriched with patch-level information, are used to generate predictions based on view-tailored text features. These predictions are then regularised across views, complementing the earlier consistency mechanisms and contributing to a cohesive overall framework. Our approach outperforms existing methods across multiple benchmarks, including base-to-novel generalisation, domain generalisation, and cross-dataset evaluation. These results underscore the potential of integrating fine-grained details for more robust and adaptable prompts, marking a step forward in foundation model tuning.

## 1 INTRODUCTION

Foundation models, such as CLIP (Radford et al., 2021), have demonstrated impressive zero-shot performance across a wide range of tasks, showcasing their adaptability and potential for general applications. However, when it comes to task-specific performance, these models often fall short, compared to specialised models. Fine-tuning foundation models to obtain task-specific models can address this gap, but it is computationally expensive, time-consuming, and impractical when we have only limited data. To overcome these limitations, prompt tuning has emerged as an efficient alternative for adapting large foundation models like CLIP (Zhou et al., 2021). This method introduces learnable prompts to make minor adjustments, while keeping the rest of the model frozen, drastically reducing the computational overhead (Zhou et al., 2021; 2022; Zhu et al., 2022; Yao et al., 2023; Khattak et al., 2022; Derakhshani et al., 2022; Khattak et al., 2023; Roy & Etemad, 2023; Zhang et al., 2023).

Despite its efficiency, prompt tuning tends to overfit on base classification task which includes a subset of base classes, and diminishes generalisation to novel, unseen tasks. Models fine-tuned with prompts can underperform on unseen tasks, compared to CLIP's original zero-shot capabilities (Zhou et al., 2022). This overfitting is due to a loss of generalisation, where the model becomes too tailored to the specific training data, limiting its broader applicability. To address this issue, various strategies have been proposed, such as generating text prompts conditioned on image features (Zhou et al., 2022; Zang et al., 2022), treating prompt learning as a Bayesian inference problem (Derakhshani et al., 2022), aligning prompt updates with gradients from non-prompted predictions (Zhu et al., 2022), and applying consistency-based regularisation to the model's output features and logits (Khattak

et al., 2023; Roy & Etemad, 2023). Among these, multi-modal frameworks that regularise output features and logits with zero-shot CLIP, using $\ell$1-norm or cosine similarity (Khattak et al., 2023; Roy & Etemad, 2023), have shown notable success in improving generalisation.

Existing methods, such as PromptSRC and CoPrompt, focus primarily on global text and vision features, overlooking the rich local information in image patches. Self-supervised frameworks (Yun et al., 2022; Li et al., 2022; Atito et al., 2022) incorporate patches but primarily optimize patch-level losses without addressing cross-modality consistency. Adapter based methods, such as CoPrompt (Roy & Etemad, 2023), also fail to capture rich patch-level details for vision feature generation. In contrast, our approach integrates patch-level features across multiple components, leveraging inter-view and intra-patch-level consistency to address finer-grained variability beyond global consistency. By using patches for both vision feature, text and logit regularization and conditioning text prompts, our method improves feature generation and class predictions, enhancing generalization.

Specifically, we introduce a novel patch-level consistency loss that enforces both intra-view and inter-view consistency, aligning patch representations across different views. While prior works focus on global consistency, we apply intra-view regularisation to the prompted patch features from the anchor view with zero-shot anchor patches. In addition to promote further consistency and to design a non-trivial regularisation, we enforce inter-view consistency which maintains patch consistency across different views of an image by adding an extra guidance for generalisable prompts. An extra augmented view is generated and patch consistency loss is applied with most similar zero-shot anchor patches. Here, both the anchor and augmented prompted patch features are projected through a convolutional layer to obtain better feature representations. This combined intra-view and inter-view consistency ensures robust prompt tuning and significantly enhances generalisation.

In addition to regularizing the vision branch, we also focus on the text branch by leveraging local information and inter-view variations. To achieve this, we generate view-tailored text prompts based on view-specific clustered patch features. Unlike CoCoop (Zhou et al., 2022), which relies on global semantics for text prompt generation, our approach incorporates local patch features for a more refined prompt. Finally, view-specific text features are generated from each of the view-tailored prompts. To further regularize the view-tailored text prompts, we apply an $\ell$1-loss between the anchor-view and augmented-view text features, with a stop-gradient applied to the anchor-view output. This ensures inter-view consistency in the text branch and promotes the generalizability of the learned text prompts.

Lastly, patch-level information is integrated with vision features by averaging projected patch tokens and combining them with class token features to produce enhanced vision outputs. These outputs are used to compute logits via a dot product with view-specific text features, further regularized with $\mathcal{KL}$-Divergence between anchor and augmented view logits, applying a stop-gradient to the anchor view.

Our **P**atch-**A**ware **P**rompting (PAP) framework achieves superior performance over previous methods across diverse tasks, including base-to-novel generalization, domain generalization, and cross-dataset evaluation. By seamlessly integrating patch-level information and addressing both intra-view and inter-view variations, PAP enhances generalization and robustness in prompt tuning. While PAP includes multiple components, each is specifically designed to address distinct challenges in leveraging patch-level information, with their collective effectiveness validated through extensive ablation studies. This ensures a reliable and efficient pipeline for adapting foundation models to downstream tasks.

## 2 RELATED WORK

Foundation models based on image-text alignment have demonstrated strong zero-shot performance, but finetuning them is computationally expensive and often leads to reduced generalisation (Radford et al., 2021; Yao et al., 2021; Jia et al., 2021; Zhai et al., 2021; Yuan et al., 2021). As a result, many applications leverage these models as feature extractors with task-specific components (Vidit et al., 2023; Yu et al., 2023; Lin et al., 2023; Yi et al., 2023; Yun et al., 2023; Bousselham et al., 2023). Prompt learning offers an efficient alternative by adapting vision-language models with task-specific prompts while keeping the base model frozen, thus avoiding the challenges of full finetuning (Zhou et al., 2021). However, prompts trained on base classes tend to generalise poorly to unseen classes. To address this, several methods apply regularisation techniques—such as loss constraints, input

conditioning, or different strategies/components (Zhou et al., 2022; Khattak et al., 2023; Roy & Etemad, 2023; Derakhshani et al., 2022)—to improve generalisation and ensure better adaptation across both base and novel classes.

CoCoop (Zhou et al., 2022), UniVL (Zang et al., 2022), and MaPLe (Khattak et al., 2022) regularise prompts by conditioning inputs with global information. CoCoop uses vision features to condition text prompts, UniVL jointly learns vision and text prompts via a small neural network, and MaPLe synchronises vision and text prompts across layers. In contrast, our method conditions text prompts at the input layer using rich patch information and regularises output features with a loss function. PromptSRC (Khattak et al., 2023) and CoPrompt (Roy & Etemad, 2023) focus on feature and prediction regularisation with zero-shot CLIP knowledge. PromptSRC regularises global embeddings and logits while aggregating prompts across epochs, whereas CoPrompt enhances feature learning with an adapter (Gao et al., 2021). Our approach emphasises finer patch-level feature regularisation and inter-view consistency. Independently developed, Long et al. (2024) uses clustered patch tokens for text prompts but lacks inter-view consistency and patch integration into predictions, underperforming compared to PromptSRC. Prograd (Zhu et al., 2022) and Bayesian Prompt (Derakhshani et al., 2022) propose different strategies for prompt regularisation. Prograd (Zhu et al., 2022) updates prompts only when their gradients align with zero-shot knowledge, while ProDA (Lu et al., 2022) enforces a Gaussian distribution on prompts to reduce bias. Bayesian Prompt (Derakhshani et al., 2022) views prompt tuning through a Bayesian lens to improve generalisation.

In contrast to these regularisation methods, recent studies focus on improving base class performance (Zhang et al., 2023) or distilling knowledge from a larger teacher model (Li et al., 2024). DePT (Zhang et al., 2023) decouples the learning of base and novel classes by adding a classification head specifically for base classes, improving base class performance while maintaining results on novel classes. PromptKD (Li et al., 2024) distills a larger CLIP-L/14 model into a smaller CLIP-B/16 model, enhancing overall performance. CasPL Wu et al. (2024), utilizing the larger CLIP-L/14 model for distilling adaptable prompts, follows a two-stage process: learning adaptable prompts via unsupervised learning and training boosting prompts using prior prompt-tuning methods. Orthogonally, MetaPrompt (Park et al., 2024) applies meta-learning for prompt tuning by learning a regulariser. Unlike these methods, we maintain the existing training pipeline, avoid distilling from larger models, and focus on improving performance through loss functions and by exploiting fine-grained details.

Our Patch-Aware Prompting (PAP) differs from existing regularisation techniques by incorporating patch-level information at multiple stages. We regularise vision embeddings at the patch level across both similar and different views for more effective regularisation. Additionally, we use patch information to generate view-specific text prompts, which are further regularised to maintain consistency across views. Finally, we enhance the vision embeddings with patch-level information while also regularising logits at the inter-view level. This design allows our model to perform competitively across several evaluation benchmarks.

## 3 METHODOLOGY

We propose a novel prompt tuning framework that leverages rich, local contextual information to maintain the generalisability of CLIP to novel tasks, while improving performance on base tasks. Our framework is a modular component that can be applied to previous global prompt consistency methods (Khattak et al., 2023; Roy & Etemad, 2023; Zhang et al., 2023). By incorporating patch-level information, we design components that further regularize the prompt learning process. Unlike earlier approaches that rely on global feature embeddings (Khattak et al., 2023; Roy & Etemad, 2023) or architectural modifications (Zhou et al., 2022; Zhu et al., 2022) to enhance generalisability, our method focuses on local information. Fig. 1 depicts the overall architecture of the proposed Patch-Aware Prompting framework.

First, we introduce a novel vision patch regularisation loss. This loss regularizes the patch outputs generated by our framework with zero-shot predictions from CLIP (Radford et al., 2021), ensuring consistency at the patch level. To handle inter-view variance among different crops typically sampled during training, we add another view called the augmented view of the image and apply a patch loss using zero-shot predictions from the anchor view. This ensures consistent representation with CLIP, both within similar views and across different views of an image. To maintain inter-view consistency at the textual level and incorporate rich local context, we generate view-specific text prompts for both

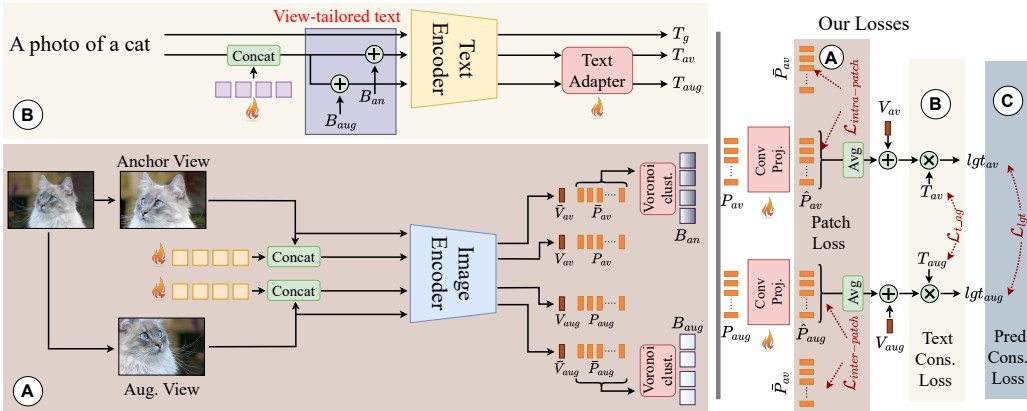

Figure 1: Patch-Aware Prompting (PAP) Framework: Patch information from anchor and augmented views is integrated into (A)vision, (B)text branches and (C)predictions. View-tailored text prompts are conditioned on patch clusters, with intra-view and inter-view losses applied to vision patches and inter-view consistency losses applied to text prompts and patch enhanced predictions.

the anchor and augmented views. This process differs from previous frameworks that rely on global image features (Zhou et al., 2021; Zang et al., 2022) by utilising patches that contain fine-grained details. We use the Voronoi algorithm (Voronoi, 1908) to cluster patch features, creating biases that guide the text prompt initialization. We further regularise these prompts using a text loss that ensures consistency between the text features generated from the anchor and the augmented prompts. This alignment across view-specific prompts improves generalisability.

Lastly, we combine the patch-level information to generate enhanced vision features. The final vision feature is a combination of the class token embedding and the average of the patch token embeddings. This ensures that patch-level information is included, providing a comprehensive representation of the image. We then compute the logits by taking the dot product of the enhanced vision and text output features. To maintain consistency not only at the feature level but also in class predictions across views, we regularize the logits using a $\mathcal{KL}$-divergence loss between the predictions from anchor and augmented views. We provide further details of our architecture in the following sections along with preliminaries.

### 3.1 PRELIMINARIES

Let $f$ and $g$ represent the image and text encoders of CLIP, respectively. The input image $\mathbf{X} \in \mathbf{R}^{CH \times H \times W}$ is divided into $M$ patch tokens after the projection layer. The vision transformer encoder receives the input $\mathbf{X}_{in} = \{e_{cls}, e_1, \ldots, e_M\}$, consisting of the class token $e_{cls}$ and the patch tokens. After passing through the vision encoder $f$, the class and patch features are $\bar{\mathbf{V}} \in \mathbf{R}^d$ and $\bar{\mathbf{P}} \in \mathbf{R}^{M \times d}$. The class label $c_K$, from the set $\{1, \ldots, C\}$, is wrapped in the text template "a photo of a class label". The input to the text encoder, after embedding and adding [SOS] and [EOS] tokens, is $\mathbf{Y}_{in} = \{y_{[SOS]}, y_1, \ldots, c_k, y_{[EOS]}\}$, where $c_k$ and $\{y_l\}_{l=1}^L$ are the embeddings for the class label and text template. The zero-shot text features for class label $k$ are $\bar{\mathbf{T}}_k \in \mathbf{R}^d$, with $\bar{\mathbf{T}} = \{\bar{\mathbf{T}}_k\}_{k=1}^K$. The zero-shot logits are calculated as $lgt = sim(\bar{\mathbf{T}}, \bar{\mathbf{V}})/\tau$, where $\tau$ is the temperature and $sim()$ denotes the cosine similarity. The prediction for class label $k$ is given by: $\frac{\exp(sim(\bar{\mathbf{T}}_k, \bar{\mathbf{V}})/\tau)}{\sum_{c=1}^C \exp(sim(\bar{\mathbf{T}}_c, \bar{\mathbf{V}})/\tau)}$.

Prompt learning involves adding learnable prompts to either the text or image encoder (Zhou et al., 2022; 2021; Roy & Etemad, 2023; Khattak et al., 2023). Let $G_t = \{g_t^1, g_t^2, \ldots, g_t^{nt}\}$ and $G_v = \{g_v^1, g_v^2, \ldots, g_v^{nv}\}$ represent the text and vision prompts, where $nt$ and $nv$ denote the number of text and vision prompts, respectively. The inputs to the vision and text encoders with these trainable prompts are $\mathbf{X}_{in}^p = \{G_v, e_{cls}, e_1, \ldots, e_M\}$ and $\mathbf{Y}_{in}^p = \{y_{[SOS]}, G_t, c_k, y_{[EOS]}\}$. In deep prompting, additional prompts are introduced at each transformer layer, replacing the previous layer's prompts. The vision-prompted class and patch features, after passing through the vision encoder, are $\mathbf{V} \in \mathbf{R}^d$ and $\mathbf{P} \in \mathbf{R}^{M \times d}$, and the text-prompted features $\mathbf{T}_k \in \mathbf{R}^d$ are generated by the text encoder. The

training is supervised using the cross-entropy loss:

$$\mathcal{L}_{ce} = -log \frac{exp(sim(T_k, V))/\tau}{\sum_{c=1}^{C} exp(sim(T_c, V))/\tau}. \tag{1}$$

Additionally, PromptSRC (Khattak et al., 2023) defines global-level losses on features and logits:

$$\mathcal{L}_{\text{SCL-image}} = \sum_{i=1}^{d} |\bar{\boldsymbol{V}} - \boldsymbol{V}|, \ \mathcal{L}_{\text{SCL-text}} = \sum_{i=1}^{d} |\bar{\boldsymbol{T}} - \boldsymbol{T}|. \tag{2}$$

$$\mathcal{L}_{\text{SCL-logits}} = \mathcal{KL}\text{-Div}(\text{sim}(\bar{\boldsymbol{T}}, \bar{\boldsymbol{V}}), \text{sim}(\boldsymbol{T}, \boldsymbol{V})). \tag{3}$$

The final loss is a combination of the above loss functions:

$$\mathcal{L}_{\text{global}} = \mathcal{L}_{\text{CE}} + \mathcal{L}_{\text{SCL-text}} + \mathcal{L}_{\text{SCL-image}} + \mathcal{L}_{\text{SCL-logits}}. \tag{4}$$

### 3.2 PATCHES FOR PROMPT TUNING

We incorporate patch information in multiple ways in both the text and vision branches of our model. Our framework generates two views: a main view, called the anchor view $\mathbf{X_{an}}$, and an augmented view $\mathbf{X_{aug}}$. Let $\bar{\mathbf{V}}_{an} \in \mathbf{R}^d$ and $\bar{\mathbf{P}}_{an} \in \mathbf{R}^{M \times d}$ represent the zero-shot class and patch features for the anchor view, and let $\bar{\mathbf{V}}_{aug} \in \mathbf{R}^d$ and $\bar{\mathbf{P}}_{aug} \in \mathbf{R}^{M \times d}$ represent the class and patch features for the augmented view. The prompted class and patch features for the anchor view are $\mathbf{V}_{an} \in \mathbf{R}^d$ and $\mathbf{P}_{an} \in \mathbf{R}^{M \times d}$, respectively, while $\mathbf{V}_{aug} \in \mathbf{R}^d$ and $\mathbf{P}_{aug} \in \mathbf{R}^{M \times d}$ correspond to the augmented view. We further project these prompted patch features through a convolution projection block to obtain final patch features: $\hat{\mathbf{P}}_{an} = \text{ConvProj}(\mathbf{P}_{an})$ and $\hat{\mathbf{P}}_{aug} = \text{ConvProj}(\mathbf{P}_{aug})$. This convolutional projection block consists of two convolutional layers with a kernel size of $3 \times 3$.

**Patch Consistency Loss**: Consistency-guided methods for prompt tuning typically focus on global feature consistency (Khattak et al., 2023; Roy & Etemad, 2023). In contrast, we propose a patch-level consistency loss that directs the learning trajectory towards better generalization, guided by the patch information. We start by introducing a simple loss to ensure consistency between patches from the prompted and zero-shot outputs for the same view. For the anchor view, we define an intra-view patch consistency loss:

$$\mathcal{L}_{\text{intra-view}} = \sum_{i=1}^{M} (1 - \text{sim}(\bar{\boldsymbol{P}}_{an}^i - \hat{\boldsymbol{P}}_{an}^i)), \tag{5}$$

which keeps the patches aligned with the original CLIP model, helping prevent catastrophic forgetting.

During training, prompt-tuning models crop random image portions. Our goal is to preserve patch relationships across crops for better regularization. CoPrompt (Roy & Etemad, 2023) enforces global inter-view consistency but ignores patch-level details. FILIP (Yao et al., 2021) focuses on image-text alignment but not patch consistency across crops. To address this, we introduce the augmented view $\mathbf{X_{aug}}$ to ensure patch-level consistency with the anchor view $\mathbf{X_{an}}$. From $\mathbf{X_{aug}}$ patches, $\hat{\mathbf{P}}_{aug}^i (i \in \{1, \dots, M\})$, we identify the closest zero-shot patch in $\mathbf{X_{an}}$. The closest patch is determined by

$$\text{crossview\_feat}(\hat{\boldsymbol{P}}_{\text{aug}}^i) = \bar{\boldsymbol{P}}_{\text{an}}^j, \qquad j = \underset{k \in \{1, \dots, M\}}{\arg\max} \ \text{sim}(\bar{\boldsymbol{P}}_{\text{aug}}^i, \bar{\boldsymbol{P}}_{\text{an}}^k), \tag{6}$$

where the closest zero-shot anchor patch $\hat{\mathbf{P}}_{an}^j (j \in \{1, \dots, M\})$ corresponds to the zero-shot augmented patch $\hat{\mathbf{P}}_{aug}^i$. Using zero-shot outputs to calculate similarity prevents the model from finding an easier learning path, such as having all prompted patches match closely with a single target patch.

The final inter-view patch consistency loss is defined as:

$$\mathcal{L}_{\text{inter-view}} = \sum_{i=1}^{M} (1 - \text{sim}(\hat{\boldsymbol{P}}_{aug}^i, \text{crossview\_feat}(\hat{\boldsymbol{P}}_{\text{aug}}^i))), \tag{7}$$

which promotes the consistency between the prompted patches of the augmented view $\hat{\mathbf{P}}_{aug}^i$ and the closest zero-shot anchor patches $\hat{\mathbf{P}}_{an}^i$. Our total patch consistency loss is the sum of both the inter-view and intra-view consistency losses:

$$\mathcal{L}_{\text{patch-con}} = \mathcal{L}_{\text{intra-view}} + \mathcal{L}_{\text{inter-view}}. \tag{8}$$

**View-tailored text prompts:** Regularizing text prompts with global vision semantics is crucial for strong performance on novel tasks across domains (Khattak et al., 2023; Roy & Etemad, 2023). We enhance this by incorporating rich patch-level details. Unique text prompts are generated for both the anchor and augmented views. Using class tokens to generate prompts, as in Zhou et al. (2022), creates overly similar prompts due to shared information in the tokens and fails to capture patch-level details. To address this, we cluster patches as cues for patch-level semantic reasoning in the prompts. Unlike the quantized bias in Bhardwaj et al. (2022), we derive prompts from patch-level details for more flexible and expressive representations. Instead of a single bias vector for all prompt vectors $G_t = \{g_t^1, g_t^2, \ldots, g_t^{nt}\}$, we generate individual bias vectors $B = \{b^1, b^2, \ldots, b^{nt}\}$. These are derived by clustering the vision zero-shot patch features $\bar{P}$ using the Voronoi algorithm (Voronoi, 1908) to create $nt$ clusters, each serving as a bias vector for a prompt vector:

$$\{b^1, b^2, \ldots, b^{nt}\} = \text{Voronoi\_Clustering}(\bar{P}). \tag{9}$$

These bias vectors are then added to the prompt vectors to generate the final prompt vectors $\hat{G}$:

$$\hat{G} = \{\hat{g}^1, \hat{g}^2, \ldots, \hat{g}^{nt}\}; \quad \hat{g}^i = g^i + b^i. \tag{10}$$

This approach provides better prompt regularization at the input level, allowing the finer details of the input image to guide the prompt learning process. Let $B_{an} = \{b_{an}^1, b_{an}^2, \ldots, b_{an}^{nt}\}$ and $B_{aug} = \{b_{aug}^1, b_{aug}^2, \ldots, b_{aug}^{nt}\}$ be the bias vectors generated from the anchor and augmented view patches, respectively. These bias vectors are added to the prompt vectors to produce view-tailored prompt vectors $\hat{G}_{an}$ and $\hat{G}_{aug}$, corresponding to the anchor and augmented views. These prompts are then concatenated with the class label, [EOS], and [SOS] tokens, and passed through the CLIP text encoder. The output text features are $T_{an}$ and $T_{aug}$, corresponding to the view-tailored text prompts for the anchor and augmented views. We further project these text features using a text adapter (Gao et al., 2021), which consists of two linear layers, to obtain $\hat{T}_{an} = \text{TextAdapter}(T_{an})$ and $\hat{T}_{aug} = \text{TextAdapter}(T_{aug})$. This projection aligns the text features with the enhanced vision features explained further. To regularize the output features, we develop a loss function that encourages similarity between the view-tailored outputs:

$$\mathcal{L}_{\text{view-tailor}} = |\text{stop\_grad}(\hat{\boldsymbol{T}_{an}}) - \hat{\boldsymbol{T}}_{aug}| \tag{11}$$

This promotes inter-view regularization in the text space, complementing the regularization introduced in the vision space, which fosters better generalization. Our view-tailored loss, defined in Equation 11, is an $\ell 1$-loss, where a stop-gradient operation is applied on $\hat{T}_{an}$ to encourage the augmented view text prompt to align more closely with the anchor text prompt.

**Improved vision features and logits regularisation:** After regularizing the text and vision branches using finer details from patches, we propose leveraging this information to enhance the vision features. The class embeddings from the vision encoder, typically used to generate class probabilities, focus on a single dominant semantic object. However, patches contain local information that can improve the overall feature representation. To achieve this, we propose averaging the patch projections to obtain aggregated patch-level information, which is then added to the class-specific vision features to generate the final vision features $\hat{V}$:

$$\hat{\boldsymbol{V}} = \boldsymbol{V} + \alpha \star \text{Avg}(\hat{\boldsymbol{P}}), \tag{12}$$

where $\alpha$ is a scaling factor for the aggregated patch features. The final logits corresponding to the prompted inputs are computed by taking the dot product of the vision and text output features. These logits are further regularized using $\mathcal{KL}$-Divergence between the logits of the anchor and augmented views, which is defined as follows:

$$\mathcal{L}_{\text{inter-logits}} = \mathcal{KL}\text{-Div}(\text{stop\_grad}(\text{softmax}(\text{sim}(\hat{\boldsymbol{T}_{an}}, \hat{\boldsymbol{V}}_{an}))), \text{softmax}(\text{sim}(\hat{\boldsymbol{T}}_{aug}, \hat{\boldsymbol{V}}_{aug}))). \tag{13}$$

This regularization reduces inter-view variance at the logit level, leading to more generalized features. Similar to the view-tailored loss on text embeddings, we apply a stop-gradient operation to the anchor logits. The final loss is the sum of all the introduced patch-level losses:

$$\mathcal{L}_{\text{final}} = \mathcal{L}_{\text{global}} + \lambda_p \star \mathcal{L}_{\text{patch-con}} + \lambda_t \star \mathcal{L}_{\text{view-tailor}} + \lambda_l \star \mathcal{L}_{\text{inter-logits}}, \tag{14}$$

which includes a global loss applied to both the anchor and augmented views, with $\lambda_p, \lambda_t, \lambda_l$ acting as scaling factors for the patch, view-tailored text, and inter-view logit losses, respectively.

Table 1: **Comparison with SOTA methods on base-to-novel tasks**. Best novel, base, HM accuracies are underlined; bold indicates improvements over the method with our components. * denotes our reproduction with official code and settings.

(a) **Average**

|  | Base | Novel | HM |
|---|---|---|---|
| CLIP | 69.34 | 74.22 | 71.70 |
| CoOp | 82.69 | 63.22 | 71.66 |
| Co-CoOp | 80.47 | 71.69 | 75.83 |
| KgCoOp | 80.73 | 73.60 | 77.00 |
| MaPLe | 82.28 | 75.14 | 78.55 |
| CoPrompt* | 83.66 | 76.34 | 79.84 |
| PromptSRC | 84.26 | 76.10 | 79.97 |
| → + Ours | **85.07** | **77.41** | **81.05** |
| DePT* | 85.15 | 75.73 | 80.16 |
| → + Ours | **85.65** | **77.29** | **81.25** |

(b) ImageNet

|  | Base | Novel | HM |
|---|---|---|---|
| CLIP | 72.43 | 68.14 | 70.22 |
| CoOp | 76.47 | 67.88 | 71.92 |
| Co-CoOp | 75.98 | 70.43 | 73.10 |
| KgCoOp | 75.83 | 69.96 | 72.78 |
| MaPLe | 76.66 | 70.54 | 73.47 |
| CoPrompt | 77.67 | 71.27 | 74.33 |
| PromptSRC | 77.60 | 70.73 | 74.01 |
| → + Ours | **78.00** | **70.96** | **74.33** |
| DePT | 78.20 | 70.27 | 74.02 |
| → + Ours | **78.26** | **70.90** | **74.39** |

(c) Caltech101

|  | Base | Novel | HM |
|---|---|---|---|
| CLIP | 96.84 | 94.00 | 95.40 |
| CoOp | 98.00 | 89.81 | 93.73 |
| Co-CoOp | 97.96 | 93.81 | 95.84 |
| KgCoOp | 97.72 | 94.39 | 96.03 |
| MaPLe | 97.74 | 94.36 | 96.02 |
| CoPrompt | 98.27 | 94.90 | 96.55 |
| PromptSRC | 98.10 | 94.03 | 96.02 |
| → + Ours | **98.50** | **94.46** | **96.35** |
| DePT | 98.60 | 93.93 | 96.20 |
| → + Ours | **98.63** | **94.30** | **96.41** |

(d) OxfordPets

|  | Base | Novel | HM |
|---|---|---|---|
| CLIP | 91.17 | 97.26 | 94.12 |
| CoOp | 93.67 | 95.29 | 94.47 |
| Co-CoOp | 95.20 | 97.69 | 96.43 |
| KgCoOp | 94.65 | 97.76 | 96.18 |
| MaPLe | 95.43 | 97.76 | 96.58 |
| CoPrompt | 95.03 | 96.86 | 95.93 |
| PromptSRC | 95.33 | 97.30 | 96.30 |
| → + Ours | **95.70** | **97.80** | **96.73** |
| DePT | 95.46 | 97.16 | 96.30 |
| → + Ours | **95.70** | **97.83** | **96.75** |

(e) StanfordCars

|  | Base | Novel | HM |
|---|---|---|---|
| CLIP | 63.37 | 74.89 | 68.65 |
| CoOp | 78.12 | 60.40 | 68.13 |
| Co-CoOp | 70.49 | 73.59 | 72.01 |
| KgCoOp | 71.76 | 75.04 | 73.36 |
| MaPLe | 72.94 | 74.00 | 73.47 |
| CoPrompt | 76.97 | 74.40 | 75.66 |
| PromptSRC | 78.27 | 74.97 | 76.58 |
| → + Ours | **80.03** | **75.30** | **77.59** |
| DePT | 80.80 | 74.76 | 77.66 |
| → + Ours | **81.53** | **75.23** | **78.25** |

(f) Flowers102

|  | Base | Novel | HM |
|---|---|---|---|
| CLIP | 72.08 | 77.80 | 74.83 |
| CoOp | 97.60 | 59.67 | 74.06 |
| Co-CoOp | 94.87 | 71.75 | 81.71 |
| KgCoOp | 95.00 | 74.73 | 83.65 |
| MaPLe | 95.92 | 72.46 | 82.56 |
| CoPrompt | 97.27 | 76.60 | 85.71 |
| PromptSRC | 98.07 | 76.50 | 85.95 |
| → + Ours | **98.33** | **77.40** | **86.61** |
| DePT | 98.60 | 76.80 | 86.34 |
| → + Ours | **98.63** | **77.40** | **86.73** |

(g) Food101

|  | Base | Novel | HM |
|---|---|---|---|
| CLIP | 90.10 | 91.22 | 90.66 |
| CoOp | 88.33 | 82.26 | 85.19 |
| Co-CoOp | 90.70 | 91.29 | 90.99 |
| KgCoOp | 90.50 | 91.70 | 91.09 |
| MaPLe | 90.71 | 92.05 | 91.38 |
| CoPrompt | 90.16 | 91.53 | 90.83 |
| PromptSRC | 90.67 | 91.53 | 91.10 |
| → + Ours | **90.83** | **92.06** | **91.44** |
| DePT | 90.80 | 91.53 | 91.16 |
| → + Ours | **90.86** | **91.93** | **91.39** |

(h) FGVCAircraft

|  | Base | Novel | HM |
|---|---|---|---|
| CLIP | 27.19 | 36.29 | 31.09 |
| CoOp | 40.44 | 22.30 | 28.75 |
| Co-CoOp | 33.41 | 23.71 | 27.74 |
| KgCoOp | 36.21 | 33.55 | 34.83 |
| MaPLe | 37.44 | 35.61 | 36.50 |
| CoPrompt | 40.20 | 39.33 | 39.76 |
| PromptSRC | 42.73 | 37.87 | 40.15 |
| → + Ours | **44.46** | **38.60** | **41.10** |
| DePT | 45.70 | 36.73 | 40.73 |
| → + Ours | **46.60** | **38.50** | **42.16** |

(i) SUN397

|  | Base | Novel | HM |
|---|---|---|---|
| CLIP | 69.36 | 75.35 | 72.23 |
| CoOp | 80.60 | 65.89 | 72.51 |
| Co-CoOp | 79.74 | 76.86 | 78.27 |
| KgCoOp | 80.29 | 76.53 | 78.36 |
| MaPLe | 80.82 | 78.70 | 79.75 |
| CoPrompt | 82.33 | 79.50 | 80.89 |
| PromptSRC | 82.67 | 78.47 | 80.52 |
| → + Ours | **83.23** | **79.10** | **81.11** |
| DePT | 83.27 | 78.97 | 81.06 |
| → + Ours | **83.40** | **79.10** | **81.19** |

(j) DTD

|  | Base | Novel | HM |
|---|---|---|---|
| CLIP | 53.24 | 59.90 | 56.37 |
| CoOp | 79.44 | 41.18 | 54.24 |
| Co-CoOp | 77.01 | 56.00 | 64.85 |
| KgCoOp | 77.55 | 54.99 | 64.35 |
| MaPLe | 80.36 | 59.18 | 68.16 |
| CoPrompt | 82.93 | 62.80 | 71.47 |
| PromptSRC | 83.37 | 62.97 | 71.75 |
| → + Ours | **84.00** | **64.63** | **73.05** |
| DePT | 84.03 | 60.16 | 70.11 |
| → + Ours | **85.06** | **64.33** | **73.25** |

(k) EuroSAT

|  | Base | Novel | HM |
|---|---|---|---|
| CLIP | 56.48 | 64.05 | 60.03 |
| CoOp | 92.19 | 54.74 | 68.69 |
| Co-CoOp | 87.49 | 60.04 | 71.21 |
| KgCoOp | 85.64 | 64.34 | 73.48 |
| MaPLe | 94.07 | 73.23 | 82.35 |
| CoPrompt | 93.20 | 73.96 | 82.40 |
| PromptSRC | 92.90 | 73.90 | 82.32 |
| → + Ours | **94.46** | **81.20** | **87.32** |
| DePT | 93.33 | 75.43 | 83.41 |
| → + Ours | **95.23** | **81.26** | **87.69** |

(l) UCF101

|  | Base | Novel | HM |
|---|---|---|---|
| CLIP | 70.53 | 77.50 | 73.85 |
| CoOp | 84.69 | 56.05 | 67.46 |
| Co-CoOp | 82.33 | 73.45 | 77.64 |
| KgCoOp | 82.89 | 76.67 | 79.65 |
| MaPLe | 83.00 | 78.66 | 80.77 |
| CoPrompt | 86.26 | 78.63 | 83.07 |
| PromptSRC | 87.10 | 78.80 | 82.74 |
| → + Ours | **87.93** | **80.06** | **83.81** |
| DePT | 87.93 | 77.33 | 82.29 |
| → + Ours | **88.33** | **79.36** | **83.60** |

# 4 EXPERIMENTS

The evaluation is done on several different tasks including base-to-novel generalization, cross-dataset evaluation and domain generalisation, with CLIP-B/16 as the frozen backbone. The protocol followed is similar to previous works like CoCoop (Zhou et al., 2022), PromptSRC (Khattak et al., 2023), CoPrompt (Roy & Etemad, 2023). For base to novel generalisation, cross dataset evaluation, we use 11 datasets namely ImageNet (Deng et al., 2009), Caltech101 (Fei-Fei et al., 2004), Oxford-Pets (Parkhi et al., 2012), StanfordCars (Krause et al., 2013), Flowers102 (Nilsback & Zisserman, 2008), Food101 (Bossard et al., 2014), FGVCAircraft (Maji et al., 2013), SUN397 (Xiao et al., 2010), UCF101 (Soomro et al., 2012), DTD (Cimpoi et al., 2014), EuroSAT (Helber et al., 2019). For domain generalisation, we evaluate on the ImageNet (Deng et al., 2009), ImageNet-A (Hendrycks et al., 2021b), ImageNet-R (Hendrycks et al., 2021a), ImageNet-Sketch (Wang et al., 2019), ImageNetV2 (Recht et al., 2019) datasets. For training we use CLIP with ViT-B/16, we set the learning rate to 0.0025 and train for a maximum of 20 epochs on base-to-novel generalisation, while maximum of 5 epochs on cross-dataset generalisation and domain generalisation. We set $\lambda_p, \lambda_t, \lambda_l$ to $1.0, 0.1, 1.0$ respectively as default but modify it for individual dataset when required. The global loss scaling factors mostly follow PromptSRC. Further details are provided in supplementary. All the ablations are conducted with our model based on PromptSRC.

## 4.1 BASE TO NOVEL GENERALIZATION

We evaluate our approach when combined with PromptSRC and DePT on the main task of base-to-novel generalization. The comparison with previous SOTA methods, including PromptSRC (Khattak et al., 2023), CoPrompt (Roy & Etemad, 2023), and DePT (Zhang et al., 2023), is shown in Table 1

Table 2: Performance on cross-dataset evaluation. Best accuracies are underlined; bold indicates improvements over the method with our components. * denotes our reproduction with official code

| | Source | Target | | | | | | | | | | |
|---|---|---|---|---|---|---|---|---|---|---|---|---|
| | ImNet | Caltech | Pets | Cars | Flowers | Food | Aircraft | SUN397 | DTD | EuroSAT | UCF | *Ave.* |
| CoOp | 71.51 | 93.70 | 89.14 | 64.51 | 68.71 | 85.30 | 18.47 | 64.15 | 41.92 | 46.39 | 66.55 | 63.88 |
| Co-CoOp | 71.02 | 94.43 | 90.14 | 65.32 | 71.88 | 86.06 | 22.94 | 67.36 | 45.73 | 45.37 | 68.21 | 65.74 |
| MaPLe | 70.72 | 93.53 | 90.49 | 65.57 | 72.23 | 86.20 | 24.74 | 67.01 | 46.49 | 48.06 | 68.69 | 66.30 |
| Bayesian Prompt | 70.93 | 93.67 | 90.63 | 65.00 | 70.90 | 86.30 | 24.93 | 67.47 | 46.10 | 45.87 | 68.67 | 65.95 |
| CoPrompt* | 70.43 | 93.60 | 90.00 | 64.80 | 70.77 | 85.80 | 23.53 | 67.30 | 43.77 | 44.50 | 67.83 | 65.19 |
| PromptSRC | 71.27 | 93.60 | 90.25 | 65.70 | 70.25 | 86.15 | 23.90 | 67.10 | 46.87 | 45.50 | 68.75 | 65.81 |
| ↪ + Ours | **72.00** | **94.03** | **90.40** | **66.03** | **71.70** | **86.66** | **24.46** | **67.50** | **47.33** | **46.83** | **69.60** | **66.45** |
| DePT | 71.60 | 93.80 | 90.13 | 66.00 | 70.93 | 86.27 | 24.30 | 67.23 | 46.60 | 45.83 | 69.10 | 66.02 |
| ↪ + Ours | **72.03** | **94.03** | **90.50** | 66.00 | **71.80** | 86.63 | **24.76** | **67.60** | **47.46** | **46.73** | **69.83** | **66.53** |

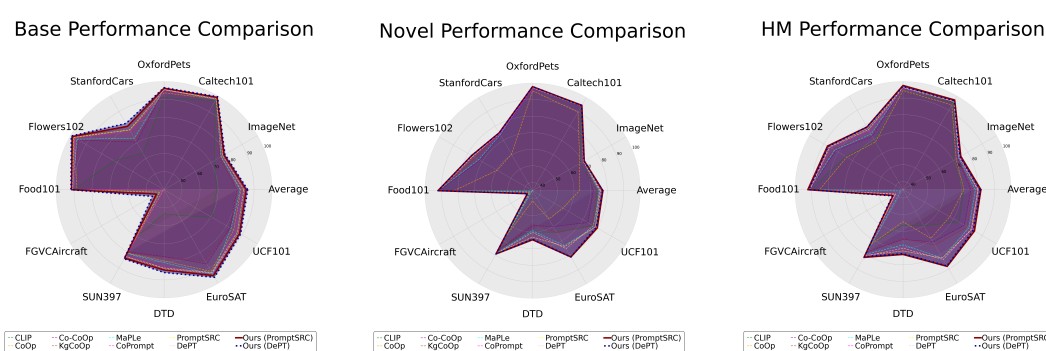

Figure 2: Performance Comparison for Base, Novel, and HM metrics across different methods.

and Figure 2. Our method consistently performs better across various datasets on novel, base, and harmonic mean (HM) when combined with both PromptSRC and DePT. On average, when our method is combined with PromptSRC, it outperforms all SOTA methods, including CoPrompt and PromptSRC, in base, novel, and HM. While we trail slightly on base classes when combined with DePT, which is mainly designed to improve base class performance, our approach impressively achieves better results across all settings when paired with DePT. Notably, when paired with PromptSRC, our method improves upon PromptSRC by $0.89\%$, $1.31\%$, and $1.08\%$ on base, novel, and HM, respectively. Compared to CoPrompt, we see improvements of $1.34\%$, $0.85\%$, and $1.08\%$ on base, novel, and HM. When compared with DePT, there is only a marginal decrease of $0.08\%$ on base classes, but an improvement of $1.68\%$ and $0.89\%$ on novel and HM, respectively. Additionally, with DePT, our method surpasses all others on base and HM, while trailing our combination with PromptSRC by only $0.12\%$ on novel. Compared to DePT alone, we show a notable improvement of $0.5\%$, $1.56\%$, and $1.09\%$ on base, novel, and HM, respectively.

## 4.2 CROSS-DATASET EVALUATION

The evaluation on cross datasets follows PromptSRC (Khattak et al., 2023). We train on ImageNet in a few-shot manner and evaluate on diverse datasets. In Table 2, the model is fine-tuned on the source dataset (ImageNet) and evaluated on 10 target datasets in a zero-shot setting. Our method combined with PromptSRC shows strong performance across most datasets, achieving the highest average accuracy of 66.45%, outperforming DePT by 0.43% and PromptSRC by 0.64%.

When combined with DePT, our method further improves the average accuracy to 66.53%, surpassing DePT alone by 0.51% and maintaining the highest performance across several datasets, including Aircraft (24.76%), SUN397 (67.60%), and DTD (47.46%). Notably, the combination with DePT also matches or exceeds the performance of PromptSRC in challenging datasets such as DTD and UCF101, highlighting its strong generalization capability. Compared to MaPLe, which achieves an average of 66.30%, our method consistently performs better, especially on datasets like Cars (66.03%) and DTD (47.46%). Overall, our method, when combined with both PromptSRC and DePT, demonstrates robust improvements, reflecting its superior generalization across diverse domains.

## 4.3 DOMAIN GENERALIZATION

The results for domain generalization are shown in Table 3. The ImageNet dataset is used as the source for fine-tuning, and the model is evaluated on four ImageNet variants. In this evaluation, the proposed method combined with PromptSRC demonstrates strong performance compared to other leading approaches. It achieves an average performance that is 0.31% higher than the base Prompt-SRC, with a notable improvement of 0.75% on ImageNetV2 and a small gap of 0.10% on ImageNetR. Similarly, when combined with DePT, the

Table 3: Domain generalization performance (best in bold).

| | Source | Target | | | | |
|---|---|---|---|---|---|---|
| | ImNet | ImNetV2 | ImNetS | ImNetA | ImNetR | Ave. |
| CLIP | 66.73 | 60.83 | 46.15 | 47.77 | 73.96 | 57.17 |
| UPT | **72.63** | 64.35 | 48.66 | 50.66 | 76.24 | 59.98 |
| CoOp | 71.51 | 64.20 | 47.99 | 49.71 | 75.21 | 59.28 |
| Co-CoOp | 71.02 | 64.07 | 48.75 | 50.63 | 76.18 | 59.90 |
| ProGrad | 72.24 | 64.73 | 47.61 | 49.39 | 74.58 | 59.07 |
| KgCoOp | 71.20 | 64.10 | 48.97 | 50.69 | 76.70 | 60.11 |
| MaPLe | 70.72 | 64.07 | 49.15 | 50.90 | 76.98 | 60.26 |
| Bayesian Prompt | 70.93 | 64.23 | 49.20 | **51.33** | 77.00 | 60.44 |
| PromptSRC | 71.27 | 64.35 | 49.55 | 50.90 | **77.80** | 60.65 |
| CoPrompt | 70.80 | 64.25 | 49.43 | 50.50 | 77.51 | 60.42 |
| Ours + PromptSRC | 72.00 | 65.10 | 50.00 | 51.03 | 77.70 | 60.96 |
| Ours + DePT | 72.03 | **65.16** | **50.03** | 51.16 | 77.70 | **61.01** |

method shows further gains, achieving an average performance that surpasses the base DePT by 0.53%, with improvements of 0.78% on ImageNetV2 and 0.14% on ImageNetR. Compared to previous SOTA, our method outperforms by 0.36%, demonstrating better generalization across varying ImageNet distributions.

Table 4: Different Components of our framework

| P. Loss | T. Text | V. Feat | Base | Novel | HM |
|---|---|---|---|---|---|
| ✗ | ✗ | ✗ | 84.55 | 75.43 | 79.89 |
| ✓ | ✗ | ✗ | 84.53 | 76.03 | 80.17 |
| ✗ | ✓ | ✗ | 84.60 | 75.79 | 80.09 |
| ✗ | ✗ | ✓ | 84.70 | 76.07 | 80.10 |
| ✓ | ✓ | ✓ | 85.07 | 77.41 | 81.05 |

Table 5: Different Losses

| $\lambda_p$ | $\lambda_t$ | $\lambda_l$ | Base | Novel | HM |
|---|---|---|---|---|---|
| ✗ | ✗ | ✗ | 83.40 | 76.90 | 79.98 |
| ✓ | ✗ | ✗ | 84.51 | 76.27 | 80.25 |
| ✗ | ✓ | ✗ | 84.67 | 76.04 | 80.20 |
| ✗ | ✗ | ✓ | 84.67 | 76.24 | 80.20 |
| ✓ | ✓ | ✓ | 85.07 | 77.41 | 81.05 |

Table 6: Patch Loss Comparison

| Intra | Inter | Base | Novel | HM |
|---|---|---|---|---|
| ✗ | ✗ | 84.49 | 76.02 | 80.06 |
| ✓ | ✗ | 84.53 | 75.71 | 80.03 |
| ✗ | ✓ | 84.31 | 75.89 | 80.01 |
| ✓ | ✓ | 85.07 | 77.41 | 81.05 |

## 4.4 ABLATIONS

**Effect of different components introduced:** We studied the effects of different components we introduced, mainly patch loss, view-tailored text, and enhanced vision features—on performance, as shown in Table 4. Each component improves performance, particularly on novel tasks. However, combining enhanced vision features, patch loss, and view-tailored text improves base, novel, and HM performance. We also evaluated the effect of different conditioning inputs to generate view-tailored text prompts, as shown in Table 7, and found that initializing prompts using patch clusters works best. CoCoop (Zhou et al., 2022), which uses only the class token, performs poorly compared to both our method and attention-based conditioning. Cross-attention from text prompts to patches performs slightly worse than our method but still better than CoCoop, highlighting the importance of patch-level conditioning.

We also studied different clustering techniques in Table 8. KMeans performed comparably to our Voronoi-based clustering on base classes but dropped significantly on novel classes, while EM was inferior to both. Voronoi clustering generates more generalizable clusters, improving performance across base and novel classes. In Table 9, we examined the effects of convolution projection and the use of adapters. We found that a text adapter is required when using convolution projection on patches, while using adapters on either text or vision alone improves performance. Our default setup, which includes both text adapters and convolution projection, performs best.

We analyzed the effects of adding small crops (scale 0.05 to 0.4) and varying the number of large crops (scale 0.05 to 1) in Table 10. Using two large crops without small crops provided the best performance, while adding small crops or more large crops led to a slight decline. In Table 12, simple augmentations for both anchor and augment views performed best on novel classes, while complex augmentations slightly decreased overall performance. The combination of simple anchor and complex augmentations remained competitive but slightly underperformed on both base and novel classes.

**Effect of different losses:** We study the effects of different losses introduced in our framework, as shown in Table 5. We find that each of the losses improves performance compared to cases where no

losses are applied. Our default case, where all losses are applied, performs the best. This demonstrates that regularization is necessary at both the feature level and the logit level to maintain the performance of our framework. Additionally, we evaluate the individual effects of intra- and inter-view patch losses in Table 6, while keeping the rest of the losses constant during the experiments. Applying either intra- or inter-view patch loss alone results in a drop in performance, particularly on novel classes. This indicates that utilizing both intra- and inter-view losses boosts base accuracy while enhancing generalization to novel classes.

Table 7: Comparison of Input Conditioning Techniques

| Method | Base | Novel | HM |
|---|---|---|---|
| CoCoop | 84.30 | 75.00 | 79.45 |
| Attention | 84.32 | 76.44 | 80.24 |
| Ours | 85.07 | 77.41 | 81.05 |

Table 8: Comparison of Clustering Techniques

| Method | Base | Novel | HM |
|---|---|---|---|
| KMeans | 84.36 | 74.97 | 79.51 |
| EM | 84.28 | 74.37 | 79.22 |
| Voronoi | 85.07 | 77.41 | 81.05 |

Table 9: Comparison of Projection & Adapter

| Text | Vision | Base | Novel | HM |
|---|---|---|---|---|
| ✗ | ✗ | 83.24 | 75.31 | 79.08 |
| ✓ | ✗ | 84.40 | 75.52 | 80.02 |
| ✗ | Adapter | 84.25 | 75.69 | 80.00 |
| ✗ | ConvProj | 84.36 | 76.07 | 80.09 |
| ✓ | Adapter | 84.21 | 75.18 | 79.61 |
| ✓ | ConvProj | 85.07 | 77.41 | 81.05 |

Table 10: Comparison of Crops

| L | S | Base | Novel | HM |
|---|---|---|---|---|
| 1 | 0 | 84.27 | 75.58 | 79.68 |
| 2 | 1 | 84.36 | 74.80 | 79.29 |
| 2 | 10 | 84.51 | 74.10 | 78.96 |
| 3 | 0 | 84.50 | 75.97 | 80.00 |
| 2 | 0 | 85.07 | 77.41 | 81.05 |

Table 11: Addition to Competing Methods

| Method | Base | Novel | HM |
|---|---|---|---|
| Cocoop | 80.47 | 71.69 | 75.83 |
| Cocoop + Our | 81.47 | 72.43 | 76.68 |
| CoPrompt | 83.66 | 76.34 | 79.84 |
| CoPrompt + Our | 84.26 | 77.40 | 80.68 |

Table 12: Comparison of Augmentation

| Anchor | Augment | Base | Novel | HM |
|---|---|---|---|---|
| Simple | Simple | 85.07 | 77.41 | 81.06 |
| Simple | Complex | 84.46 | 76.31 | 80.18 |
| Complex | Simple | 84.38 | 76.14 | 80.05 |
| Complex | Complex | 83.71 | 74.87 | 79.04 |

**Applicability to different methods:** We apply our framework to both CoCoop (Zhou et al., 2022) and CoPrompt (Roy & Etemad, 2023), as shown in Table 11. Our method improves CoCoop by 1.00% on base, 0.74% on novel, and 0.85% on HM. Similarly, it improves CoPrompt by 0.60% on base, 1.06% on novel, and 0.84% on HM. These improvements highlight the effectiveness of incorporating patch-level information for enhancing performance.

## 4.5 RUNTIME & MEMORY CONSUMPTION

Table 13 compares the learnable parameters, GPU memory consumption, and training time for various methods trained on Food101 for 10 epochs. While our method introduces a slight increase in learnable parameters—4.89M for PromptSRC and 5.12M for DePT, this increase remains minimal when compared to the default CLIP model. The use of view/image-tailored prompts, similar to CoCoop, and having multiple views results in a modest rise in GPU memory usage (2.42G) and a training time of 13 min 47 sec. However, unlike CoCoop and other baseline methods,

Table 13: Comparison of Total and Learnable Parameters for Different Methods

| Model | Total P. | Learnable P. | G.Mem | Train. Time |
|---|---|---|---|---|
| CLIP (ViT-B/16) | 149.62M | - | 1.35G | - |
| CoCoop | 153.15M | 3.53M | 1.24G | 13:07 Min |
| CoPrompt | 154.62M | 4.74M | 1.84G | 06:20 Min |
| PromptSRC | 150.08M | 0.46M | 2.07G | 06:06 Min |
| ↪ + Ours | 154.51M | 4.89M | 2.42G | 13:47 Min |
| DePT | 150.36M | 0.74M | 2.07G | 06:03 Min |
| ↪ + Ours | 154.74M | 5.12M | 2.42G | 13:47 Min |

our approach achieves significantly superior performance across the board. The trade-off in computational resources is well-justified by the substantial performance improvements, demonstrating that our method balances resource use with impressive gains in performance, setting a new benchmark.

## 5 CONCLUSION

We propose a novel prompting framework for vision-language models that incorporates patch-level information across multiple components, improving both task-specific performance and generalization to novel scenarios. By utilizing patch-level loss for intra- and inter-view consistency, generating regularized view-specific text prompts, and maintaining inter-view consistency of logits, our method addresses the overfitting challenges typical of traditional prompt-tuning approaches. Extensive experiments on tasks such as base-to-novel generalization, domain adaptation, and cross-dataset evaluation demonstrate superior performance, surpassing prior methods. Additionally, ablation studies confirm the effectiveness of each component and explore alternative designs. We believe this approach will significantly advance the tuning of foundation models across diverse applications.

ETHICS STATEMENT

Our research adheres to the ICLR Code of Ethics, prioritizing fairness, privacy, and transparency. All data used is anonymized, and we carefully considered potential societal impacts, aiming to avoid harm or bias. The research fosters reproducibility, and we are committed to addressing ethical concerns responsibly, ensuring a positive contribution to the broader community.

REPRODUCIBILITY STATEMENT

Our research prioritizes reproducibility by ensuring that all experiments and results can be independently verified. Upon acceptance of this paper, we will provide the full source code, including scripts for data preprocessing, model training, and evaluation. All datasets, and relevant configurations are described in detail within the paper to allow for precise reproduction of our results. We welcome feedback to ensure transparency and reproducibility in our findings.

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

## A  ADDITIONAL IMPLEMENTATION DETAILS

For our main results we have combined our method with DePT (Zhang et al., 2023), PromptSRC (Khattak et al., 2023). The DePT model we used is the one based on PromptSRC. The experiments are conducted for different seeds and mean reported. We use a default learning rate of $0.0025$ and CLIP-B/16 as the backone. We set the scaling factors for $\mathcal{L}_{\text{SCL-text}}, \mathcal{L}_{\text{SCL-image}}, \mathcal{L}_{\text{SCL-logits}}$ similar to PromptSRC for both our combination with PromptSRC, DePT. DePT uses an additional classification head on top of PromptSRC for improving base task performance. The combination of method to DePT is very straightforward for PromptSRC. For DePT, we also apply the base classification head where inputs to this head are features before convolution projection and text adapter. We use mostly a single RTX-3090 GPU for our experiments except while performing ImageNet training for base-to-novel generalisation, domain generalisation, and cross-dataset evaluation where we use $4$ V100 GPUs. The Gaussian aggregation mean and variance for prompts is also similar to PromptSRC. The training is done for 20 epochs for base-to-novel generalisation with variation for some datasets.

## B  ABLATION FOR LOSS SCALING FACTORS

Table 14 shows the effect that loss scaling factors $\lambda_p$, $\lambda_t$, $\lambda_l$ have on our framework. Our method works best when the loss scaling factors are set depending on dataset as highlighted in Table 14. Each dataset has different sensitivity to different losses which results in having to set these factors for each individual dataset.

Table 14: Different Losses

| $\lambda_p$ | $\lambda_t$ | $\lambda_l$ | Base | Novel | HM |
|---|---|---|---|---|---|
| 0.1 | 0 | 0 | 84.56 | 75.59 | 79.82 |
| 0 | 0.1 | 0 | 84.64 | 75.55 | 79.84 |
| 0 | 0 | 0.1 | 84.70 | 76.07 | 80.15 |
| 0.1 | 0.1 | 0.1 | 84.58 | 75.51 | 79.79 |
| 1 | 0 | 0 | 84.45 | 75.36 | 79.65 |
| 0 | 1 | 0 | 84.65 | 75.88 | 80.00 |
| 0 | 0 | 1 | 84.75 | 76.29 | 80.38 |
| 1 | 1 | 1 | 84.48 | 76.16 | 80.12 |
| 2 | 0 | 0 | 84.39 | 76.34 | 80.17 |
| 0 | 2 | 0 | 84.68 | 75.68 | 80.03 |
| 0 | 0 | 2 | 84.70 | 76.02 | 80.24 |
| 2 | 2 | 2 | 84.37 | 76.30 | 80.13 |
| ✓ | ✓ | ✓ | 85.07 | 77.41 | 81.05 |

## C  ABLATION FOR PARAMETER COMPLEXITY:

We conduct experiments in Table  by training two models based on PromptSRC, both having the same parameter count of 4.89M, to ensure a fair comparison. The first model employs PromptSRC without any additional modifications, while the second integrates our proposed components: intra- and inter-view patch losses, view-tailored text with consistency loss, and cross-view prediction consistency. As shown in Table 15, the model enhanced with our components achieves superior performance across all metrics. Specifically, it improves Base accuracy from 84.55% to 85.07%, Novel accuracy from 75.43% to 77.41%, and the harmonic mean (HM) from 79.89% to 81.05%, demonstrating the effectiveness of our approach.

Table 15: Performance comparison of PromptSRC-based models with and without our components, keeping the parameter size fixed at 4.89M.

| Components | Base (%) | Novel (%) | HM (%) |
|---|---|---|---|
| No Components (PromptSRC) | 84.55 | 75.43 | 79.89 |
| Our Components | **85.07** | **77.41** | **81.05** |

## D  ABLATION FOR VISION COMPONENTS

:

Table 16 summarizes the performance improvements achieved by incrementally applying our proposed vision components. Starting with the baseline PromptSRC which has single anchor view, we observe a steady increase in performance metrics as additional components are introduced. Specifically, the introduction of the intra-view patch loss leads to a 0.35% improvement in the HM metric, while incorporating the augmented view further increases performance by 0.03% on base. The most significant enhancement is achieved with the inclusion of the inter-view patch loss, resulting in a 1.37% HM improvement over the baseline.

Table 16: Performance comparison with the incremental application of proposed components.

| Method | Base | Novel | HM |
|---|---|---|---|
| PromptSRC (anchor view) | 84.27 | 75.58 | 79.68 |
| ↪ + Intra-view patch loss | 84.53 | 75.71 | 80.03 |
| ↪ + Augmented view | 84.49 | 76.02 | 80.06 |
| ↪ + Inter-view patch loss | 85.07 | 77.41 | 81.05 |

## E  ABLATION FOR TEXT COMPONENTS:

The results presented in Table 17 highlight the effectiveness of our text-level component design. Starting from the baseline PromptSRC, introducing view-tailored text conditioned using CLS tokens Zhou et al. (2022) shows a modest improvement in the harmonic mean (HM), reflecting the benefit of incorporating view-specific information for generating text features. Replacing CLS tokens with patches to condition view-tailored text further enhances performance, demonstrating the importance of leveraging fine-grained patch information. Finally, the addition of the consistency loss results in the highest performance across all metrics, emphasizing the critical role of enforcing alignment across view-specific text prompts to improve generalization.

Table 17: Performance comparison with the incremental application of text-level components.

| Method | Base | Novel | HM |
|---|---|---|---|
| PromptSRC | 84.27 | 75.58 | 79.68 |
| PromptSRC + View-tailored text (CLS token) | 84.40 | 76.00 | 80.15 |
| PromptSRC + View-tailored text (patches) | 84.80 | 76.90 | 80.80 |
| PromptSRC + View-tailored text (patches) + Consistency loss | **85.07** | **77.41** | **81.05** |

## F  ABLATION FOR PREDICTION COMPONENTS:

The results in Table 18 demonstrate the impact of our prediction-level component design. Starting from the baseline PromptSRC, adding TextAdapter yields a modest improvement in the harmonic mean (HM), highlighting the benefit of adaptive text-driven adjustments for better predictions. Introducing ConvProj further enhances performance, showcasing the importance of projection layers in refining predictions through improved vision features. Finally, the inclusion of the consistency loss results in the highest performance across all metrics, emphasizing its critical role in aligning predictions generated from augmented and anchor features.

Table 18: Performance comparison with the incremental application of prediction-level components.

| Method | Base | Novel | HM |
|---|---|---|---|
| PromptSRC | 84.27 | 75.58 | 79.68 |
| + TextAdapter | 84.30 | 76.00 | 80.12 |
| + ConvProj | 84.85 | 76.80 | 80.72 |
| + Cross-view Consistency loss | **85.07** | **77.41** | **81.05** |

