# OpenReview forum: "A PATCH LEVEL PERSPECTIVE OF PROMPT TUNING"
_ICLR.cc/2025/Conference — Submitted to ICLR 2025_

### Official Review · Reviewer_u15T · 2024-10-30

**Soundness:** 3
**Presentation:** 3
**Contribution:** 2
**Rating:** 3
**Confidence:** 4

**Summary:**

This work aims at improving the generalization ability of existing prompt tuning methods for vision-language models.
The authors recognize that previous works often overlook the rich fine-grained local information while regularizing the learning
target. To address this issue, they introduce three consistency loss based on patch-level information into the overall objective function. By experiments under the base-to-new generalisation, cross-dataset and domain generalisation settings,
they successfully validate the effectiveness of their method.

**Strengths:**

1. Most parts of the paper are clearly presented.
2. Promising results with extensive experiments.

**Weaknesses:**

My concern focuses on the novelty of this work. This work combines consistency regularization and fine-grained information, both of which are ideas similar to those found in existing works:
* Shuvendu Roy and Ali Etemad. Consistency-guided prompt learning for vision-language models.
ArXiv, abs/2306.01195, 2023.
* Muhammad Uzair Khattak, Syed Talal Wasim, Muzammal Naseer, Salman Khan, Ming-Hsuan Yang, Fahad Shahbaz Khan:
Self-regulating Prompts: Foundational Model Adaptation without Forgetting. ICCV 2023: 15144-15154
* Long, S., Zhao, Z., Yuan, J. et al. Mutual Prompt Leaning for Vision Language Models. Int J Comput Vis (2024).

The designed losses in Eq(5),(11),(13) are simple extension to equations in PromptSRC. Eq(12) can be found in some Adapter-based methods, and Eq(6)(7) can be found similar form in contrastive pre-training methods, such as "Lewei Yao, Runhui Huang, Lu Hou, Guansong Lu, Minzhe Niu, Hang Xu, Xiaodan Liang, Zhenguo Li, Xin Jiang, Chunjing Xu: FILIP: Fine-grained Interactive Language-Image Pre-Training. ICLR 2022". The quantitative prompts in Eq(9)(10) are similar to those in "Rishabh Bhardwaj, Amrita Saha, Steven C. H. Hoi, Soujanya Poria: Vector-Quantized Input-Contextualized Soft Prompts for Natural Language Understanding. EMNLP 2022: 6776-6791". Due to the reasons mentioned above, this work is more like an incremental improvement.

Besides, the proposed method introduces too many extra hyper-parameters, and the authors mentioned that the performance on each dataset is sensitive to the setting of these hyper-parameters. As such, I am wondering how to set these hyperparameters? Also, the training time has also increased significantly.

**Questions:**

Clarity:
The use of symbols in the paper is somewhat confusing and many of them are not enough explained, although their meaning can be derived from the context in most cases. Fig 1 and the descriptions in main text are mis-matched. The description of the motivation in Eq(9)(10) is too simplified. Some of the equations are wrong, such as Eq(1)(2).

Reproducibility: This paper would have good reproducibility as most parts are simple to implement. It would be nice to supply where the Voronoi-based clustering code can be found, further improving the reproducibility.

---

> ### Author Response · Authors · 2024-11-18
> **Part(1/2)**
>
> Thank you for your valuable feedback.
>
> None of the referenced works or losses explicitly target the integration of patch-level information or the alignment of consistency across views in a multi-modal context for VL prompting, both of which are central to our method. Our architecture introduces fundamental advancements by integrating patch-level relationships and inter-view consistency across vision, text, and predictions, establishing a robust multi-level alignment.
>
> - **Vision**: Unlike PromptSRC, which focuses on global consistency and does not consider patch-level relationships, our method introduces intra-view regularization to align patches within the anchor view. To address variability across views, we add an inter-view consistency loss to align patches between a newly introduced augmented view and zero-shot anchor views, enabling superior generalization.
>
> - **Text**: For inter-view consistency in text, we generate view-specific prompts for the augmented and anchor views, conditioned on patch clusters. This approach ensures sensitivity to finer image details in the text branch, further reinforced by a cross-view text consistency loss.
>
> - **Prediction Consistency**: For holistic alignment, we combine patch features with the class token to enrich the vision representation before generating predictions. A KL-divergence loss regularizes these predictions, reinforcing consistency across views and complementing the alignment of vision and text features.
>
> Neither PromptSRC nor CoPrompt incorporates patch-level information or addresses inter-view consistency. [1], which was independently developed and released publically on September 26, 2024, just days before the ICLR deadline, could not be incorporated or referenced in our work. While [1] uses clustered patch tokens for text prompts, it relies on a single view and does not account for inter-view variances by maintaining consistency across anchor and augmented views. Additionally, it does not integrate patch information into both text and predictions and underperforms compared to PromptSRC.
>
> The losses described in our method are purpose-built for multi-modal prompting and are distinct from those in referenced works. For instance:
> - **Eq(5), (11), and (13)** maintain inter-view text, vision and logit along with intra-patch-level consistency, addressing finer-grained variability beyond the global consistency discussed in [5].
>    - **Eq (5):** $$(\mathcal{L_{\text{intra-view}}} = \sum_{i=1}^{M}(1-\operatorname{sim}(\bar{P}^i_{an} - \hat{P}^i_{an})))$$
> This differs from [5], where we use patch-level representations ($P_{an}$) compared to class tokens ($V_{an}$), as employed in [5].
>     -  **Eq (11):** $$(\mathcal{L_{\text{view-tailor}}} = | \mathrm{stop\_grad}(\hat{T}\_{an}) - \hat{T}\_{aug} |)$$  This differs from [5], where we regulate cross-view text features generated from two different views, $\hat{T}\_{an}$ and $\hat{T}\_{aug}$. In contrast, [5] applies the regulation to a text feature conditioned without image information, $T$, and its corresponding zero-shot text feature, $T\_{\text{zero-shot}}$.
>     - **Eq (13):**   $$(\mathcal{L_{\text{inter-logits}}} = \mathcal{KL\text{-Div}}( \mathrm{stop\_grad}(\mathrm{softmax}(\mathrm{sim}({\hat{T}\_{an}}, {\hat{V}\_{an}}))), \mathrm{softmax}(\mathrm{sim}({\hat{T}\_{aug}}, {\hat{V}\_{aug}}))))$$  This equation is applied to predictions generated from the anchor view and the augmented view. The purpose of this loss is to ensure that predictions, which are sensitive to fine-grained features from two different views, learn to attend to similar fine-grained concepts. This is in contrast to [5], which applies this loss to make prompt predictions, $\mathrm{sim}({T\_{prompt}}, {V\_{prompt}})$, similar to zero-shot predictions, $\mathrm{sim}({T\_{zero}}, {V\_{zero}})$, while focusing only on global concept learning.
>
>
> - **Eq (12):** $$\hat{V} = V + \alpha \star \operatorname{Avg}(\hat{P})$$ captures rich patch-level details that are often overlooked in adapter-based methods, such as those in [2]. Specifically, we use patches $\hat{P}$, enriched through a convolutional projection, to enhance the class tokens $V$. This approach ensures that the final vision features $\hat{V}$ become more sensitive to patch-level information. In contrast, adapter-based methods [2] focus on adapting class token features $V$ through a linear projection, which is  to improve global semantics.

---

> ### Author Response · Authors · 2024-11-21
> **Part(2/2)**
>
> - **Eq(6)** $$\mathrm{crossview-feat}(\hat{P}^i_{\mathrm{aug}}) = \bar{P}^j_{\mathrm{an}}, \quad j = \arg\max_{k \in \{1, \dots, M\}} \mathrm{sim}(\hat{P}^i_{\mathrm{aug}}, \bar{P}^k_{\mathrm{an}})$$ and **Eq(7)** $$\mathcal{L_{\mathrm{inter-view}}} = \sum_{i=1}^{M} \big( 1 - \mathrm{sim}(\hat{P}^i_{\mathrm{aug}}, \mathrm{crossview-feat}(\hat{P}^i_{\mathrm{aug}})) \big)$$ differ from FILIP [3], which focuses on global image-text alignment and does not account for inter-view patch consistency between different image crops, text, and predictions. Specifically, we apply **Eq(6)** and **Eq(7)** to bring the prompted patch tokens $\hat{P}\_{an}$ closer to the zero-shot anchor patches $\bar{P}\_{an}$. This enforces patch consistency across views. In contrast, in [3], for a given global vision embedding $V\_i$, a text $T\_j$ other than the corresponding text $T\_i$ is considered a positive pair for image-text pretraining. However, they do not account for patch-level relationships across different views, nor do they consider the intricacies of prompt tuning.
> - **Eq(9)** $$\{b^1, b^2, \dots, b^{nt}\} = \mathrm{Voronoi-Clustering}(\bar{P})$$ and **Eq(10)** $$\hat{G} = \{\hat{g}^1, \hat{g}^2, \dots, \hat{g}^{nt}\}; \quad \hat{g}^i = g^i + b^i$$ derive prompts from rich patch-level details, unlike the quantized bias used in [4], achieving more flexible and expressive representations. Specifically, **Eq(9)** and **Eq(10)** use Voronoi clustering of patches $\bar{P}$ to generate bias. In contrast, [4] uses a quantized EMA codebook for NLP tasks and does not consider vision information, particularly patches, to generate biases.
>
> This deliberate design bridges a critical gap by addressing patch-level variability and multi-modal inter-view consistency, enabling our method to effectively tackle fine-grained variability while ensuring robust generalization. Our approach distinguishes itself from incremental improvements or adaptations of existing methods, making it both fundamentally novel and impactful.
>
>
> 3. The hyperparameters $\lambda_p, \lambda_1, \lambda_t$ were selected using a methodology similar to [2], as partially detailed in Table 14. Despite the added complexity introduced by view-tailored text prompts, the training time remains comparable to CoCoOp (generates a different prompt for each image) while achieving significantly improved performance. Specifically, our method delivers improvements of 5.18\%, 5.72\%, and 5.42\% on base, novel, and HM metrics, respectively, for the base-to-novel task. These notable gains in performance and generalization highlight the effectiveness and efficiency of our approach.
>
> 4. The descriptions for Eq(9), Eq(10), have been incorporated in view-specific prompts para. Additionally, for improved clarity, we welcome specific examples where the use of symbols was unclear, and a new Figure 1 with more detailed caption has been provided. It is worth noting that Eq(1) and Eq(2) are directly taken from PromptSRC [5] as part of the preliminaries.
>
>
>
> [1] Mutual Prompt Leaning for Vision Language Models. Int J Comput Vis (2024)
> [2] Consistency-guided prompt learning for vision-language models. ArXiv, abs/2306.01195, 2023.
> [3] FILIP: Fine-grained Interactive Language-Image Pre-Training. ICLR 2022
> [4] Vector-Quantized Input-Contextualized Soft Prompts for Natural Language Understanding
> [5] Self-regulating Prompts: Foundational Model Adaptation without Forgetting
>
> We have updated our paper to further refine it:
>
> - We clarified the differences with PromptSRC and CoPrompt in the third paragraph of the introduction.
> - The related works section now includes [1], highlighting how it differs from our approach.
> - Additionally, we updated the relevant methodology paragraphs for Equations (6), (7), (9), and (10) to emphasize the key differences from [3] and [4].
> - We are planning to provide the whole code including vornoi clustering and other parameters used after acceptance.

---

### Official Review · Reviewer_63du · 2024-11-03

**Soundness:** 2
**Presentation:** 3
**Contribution:** 2
**Rating:** 6
**Confidence:** 5

**Summary:**

This paper introduces a new approach of prompt tuning in vision-language models, called Patch-Aware Prompting (PAP) framework, created on top of baseline model PromptSRC [1] and DePT [2]. It integrates patch level visual information, with the textual embeddings for better vision-text alignment. This patch level information are extracted through Voronoi clustering of patch features, while connected with several consistency losses. While being attached with PromptSRC and DePT, the proposed method has achieved better performances and SOTA methods, in three different generalization tasks.


[1] Ji Zhang, Shihan Wu, Lianli Gao, Hengtao Shen, and Jingkuan Song. Dept: Decoupled prompt tuning. ArXiv, abs/2309.07439, 2023.

[2] Muhammad Uzair Khattak, Syed Talal Wasim, Muzammal Naseer, Salman Siddique Khan, Ming Yang, and Fahad Shahbaz Khan. Self-regulating prompts: Foundational model adaptation without forgetting. 2023 IEEE/CVF International Conference on Computer Vision (ICCV).

**Strengths:**

1. The proposed method has used a very old clustering method called Voronoi clustering, which is interesting.
2. The paper writing and presentation are overall good.

**Weaknesses:**

1. The proposed method looks like an extension of the baseline model PromptSRC, compromising a superior level of novelty. But it is also unclear how the authors take DePT model as an independent model, where DePT is itself an augmentation approach of the existing VLM baselines.

2. Why and how is the Voronoi clustering enhancing the alignment of text and vision? Why not direct addition of class features into the text embeddings just like CoCoOp? Why the patch level features are working better than class features here? I don't want any experimental data, but concrete qualitative reasons are needed.

3. Why only PromptSRC+PAP and DePT+PAP? PAP can also be attached with CoPrompt, MaPLe, CoOp, KgCoOp etc.

4. It is recommended to consider the baselines consistent in every task, as some baselines are missing in table 1 and 2. The results of baselines used in table 3 are needed in table 1 & 2.

5. There is a mistake in Figure 1. B_{an} is denoted there as B_{av}.

**Questions:**

See the weakness section. I would like to increase my rating, if the proper justification of my questions will be given.

---

> ### Author Response · Authors · 2024-11-13
>
> Thank you for your valuable feedback.
>
>
> 1. Our method introduces a distinctive novelty by going beyond existing global models like PromptSRC and DePT, focusing on integrating patch-level information and enforcing inter-view consistency across vision, text, and prediction levels. Patch integration is critical because, while CLIP provides globally consistent representations across image crops, patch-level information varies significantly. Ensuring that patches within a view align with CLIP’s pretrained knowledge through an intra-view patch loss adds regularization but is insufficient for generalization across diverse datasets. Leveraging patch relationships across crops is essential for promoting consistency, enabling improved generalization across domains (inter-view patch loss). This further allows us to propagate similar cross-view consistency to text features and predictions, making them sensitive to patch-level variations. A summary of novel components is below:
>
> - Vision Consistency: We extend PromptSRC’s global consistency loss to patches within the anchor view, introducing intra-view regularization as an initial step. To enable generalization across diverse views, we incorporate an inter-view consistency loss between a newly introduced augmented and zero-shot anchor views. By regularizing these inter-view patch relations, patch representations are aligned more closely with zero-shot CLIP outputs, reinforcing cross-view consistency for robust generalization.
>
> - Text Consistency: To align text features across views, we generate view-specific prompts for the augmented and anchor views and introduce a consistency loss to ensure alignment between them. Since both views originate from the same image, relying on a single global class token would lead to overly similar prompts and trivial solutions. To address this, we derive unique biases for each view from clustered patches, encouraging meaningful divergence between prompts while maintaining robust alignment.
>
> - Prediction Consistency: To achieve cohesive inter-view alignment, we integrate patch features with the class token to enrich vision representations before generating predictions. This integration captures nuanced details across views, leading to more expressive predictions. A KL-divergence loss is then applied to regularize predictions across views, complementing the consistency established in vision and text levels.
>
> 2. Regarding DePT, it introduces a classification head alongside the standard predictions based on text-vision similarity. In our approach, when integrating DePT, we incorporate this classification head during training while maintaining all other components of our method.
> 3. Kindly refer to the text consistency explanation under Point 1 above for the reasoning behind Voronoi clustering.
> 3. We have provided results for our method in Table 11 and provided it below, demonstrating its effectiveness when combined with CoCoop and CoPrompt, where it further improves their performance. Additionally, KgCoCoop and CoCoop become almost identical when our method is applied, while CoPrompt is an extension of MaPLE.
>
> | Method           | Base   | Novel  | HM     |
> |------------------|--------|--------|--------|
> | Cocoop           | 80.47  | 71.69  | 75.83  |
> | Cocoop + Ours    | **81.47** | **72.43** | **76.68** |
> | CoPrompt         | 83.66  | 76.34  | 79.84  |
> | CoPrompt + Ours  | **84.26** | **77.40** | **80.68** |
>
> 4. We did not include Bayesian Prompt in Table 1 due to space constraints, as it does not outperform our method in any of the 11 subtables. For Table 3 on domain generalization, DePT does not provide results. Similarly, KgCoop and ProGrad are excluded from Tables 1 and 2 due to space limitations and their inferior performance compared to our method.
> 5. We have fixed the issue in Figure 1 in our updated manuscript.

---

> ### Author Response · Authors · 2024-11-21
>
> We have updated our paper to further refine it and address your concerns:
>
> - We provided a clearer explanation of the differences with PromptSRC and CoPrompt in the third paragraph of the introduction.
> - We improved the explanation of clustering for bias in the view-tailored text paragraphs.
> - For DePT, we added more details about its integration in the "Additional Implementation Details" section of the appendix.

---

> ### Comment · Reviewer_63du · 2024-11-27
>
> Thank you to the authors for detailed rebuttal. The given clarifications are satisfactory. I would like to increase my score to 6.

---

### Official Review · Reviewer_P4bj · 2024-11-06

**Soundness:** 3
**Presentation:** 2
**Contribution:** 2
**Rating:** 5
**Confidence:** 4

**Summary:**

This paper addresses the overfitting problem in prompt tuning for CLIP on downstream vision tasks. The authors propose several regularization methods based on patch-level consistency losses, including a patch-con loss that considers both visual inter- and intra-view consistency, a text-consistency loss across views, and an inter-logit loss between decision.

**Strengths:**

* The paper introduces regularization losses based on patch information, which serves as a valuable complement to global information.

**Weaknesses:**

* The proposed method is complex, incorporating three additional loss functions, which could make implementation and analysis challenging.
* The figure illustrating the method lacks clarity and does not sufficiently explain the approach on its own.
* The improvements demonstrated are marginal compared to recent works, such as Cascade Prompt Learning ([1]), which tackle the same issue of overfitting in prompt tuning.

Reference [1] Cascade Prompt Learning, https://arxiv.org/pdf/2409.17805

**Questions:**

* Equation 5 appears to contain a typo.
* Equation 9 is unclear: it references a "Voronoi Clustering" result as a set, which typically lacks an inherent order. How is the clustering result ordered here?

---

> ### Author Response · Authors · 2024-11-18
> **Part(1/2)**
>
> Thank you for your valuable insights. We appreciate your attention to detail and would like to clarify few details.
>
> 1. We appreciate the reviewer’s observation regarding the additional loss functions. Each loss function is purposefully designed to address specific challenges associated with integrating patch-level information and enforcing inter-view consistency across vision, text, and predictions. These losses ensure the method achieves a level of generalization and robustness that previous approaches fail to deliver.
>
> - Vision consistency: Intra-view patch loss regularizes consistency within similar views, while inter-view patch loss enforces relationships across different views. Intra-view consistency adds an additional layer of regularization by aligning with zero-shot CLIP patches. While CLIP is trained to produce consistent global representations across crops, patch-level information can vary significantly. The inter-view patch loss addresses this variability by preserving relationships between patches across views, ensuring closer alignment with zero-shot CLIP outputs and enhancing generalizability. This inter-view consistency is further extended to the text branch and used to generate predictions.
>
> - Text consistency: View-specific text prompts are initialized using patch-level information, and a text prompt consistency loss ensures coherence across views.
>
> - Prediction consistency: Predictions from vision and text features are regularized to cohesively maintain inter-view alignment.
>
>     Previous methods fail to utilize this critical patch-level information, focusing solely on patches for vision branches while ignoring patch relationships across views. This approach is insufficient for improving generalizability, as our extensive ablation studies clearly demonstrate. Each loss component is validated, showing that removing any one significantly impacts performance.
>
>     Additionally, the method’s modular design ensures seamless integration into standard training pipelines. To facilitate accessibility and adoption, we will release our implementation upon acceptance, enabling the community to replicate our results and leverage these carefully designed components to enhance generalization and robustness effectively.
>
> 2. To provide experimental justification, we have showcased the following component-level design choices. The results below compare a model with a similar parameter count to ours but trained without incorporating our proposed components. These findings demonstrate that merely increasing the number of parameters does not lead to improved performance, underscoring the critical value of our approach.
>
> | **Components**    | **Base** | **Novel** | **HM**    |
> |--------------------|----------|-----------|-----------|
> | No Components      | 84.55    | 75.43     | 79.89     |
> | Our Components     | **85.07** | **77.41** | **81.05** |
>
> We further show individual vision, text, and prediction-level component design choices justification at the experimental level. The results below demonstrate the incremental benefits of our **vision-level component design**. Starting from the baseline PromptSRC (anchor view), each proposed component contributes to consistent performance improvements. The inter-view patch loss achieves the most significant improvement, highlighting its critical role in enhancing inter-view consistency and overall performance.
>
> | **Method**                              | **Base** | **Novel** | **HM**    |
> |-----------------------------------------|----------|-----------|-----------|
> | PromptSRC (anchor view)                 | 84.27    | 75.58     | 79.68     |
> | $\hookrightarrow$ + Intra-view patch loss | 84.53    | 75.71     | 80.03     |
> | $\hookrightarrow$ + Augmented view    | 84.49    | 76.02     | 80.06     |
> | $\hookrightarrow$ + Inter-view patch loss | **85.07** | **77.41** | **81.05** |
>
> The results below demonstrate the impact of our **text-level component design**. Introducing view-tailored text conditioned on CLS tokens provides a modest improvement, while conditioning on patches further enhances performance. Adding the consistency loss achieves the best results across all metrics, highlighting the importance of enforcing alignment across view-specific text prompts.
>
> | **Method**                                            | **Base** | **Novel** | **HM**    |
> |-------------------------------------------------------|----------|-----------|-----------|
> | PromptSRC                                             | 84.27    | 75.58     | 79.68     |
> | PromptSRC + View-tailored text (CLS token)           | 84.40    | 76.00     | 80.15     |
> | PromptSRC + View-tailored text (patches)             | 84.80    | 76.90     | 80.80     |
> | PromptSRC + View-tailored text (patches) + Consistency loss | **85.07** | **77.41** | **81.05** |

---

> ### Author Response · Authors · 2024-11-21
> **Part(2/2)**
>
> The results below demonstrate the effectiveness of our **prediction-level component design**. Adding TextAdapter provides a modest improvement by leveraging adaptive text-driven features, while ConvProj further enhances predictions by refining vision representations. Finally, the consistency loss achieves the best performance across all metrics, highlighting its critical role in aligning predictions across views, generated from augmented and anchor features.
>
> | **Method**                                | **Base** | **Novel** | **HM**    |
> |-------------------------------------------|----------|-----------|-----------|
> | PromptSRC                                 | 84.27    | 75.58     | 79.68     |
> | + TextAdapter                             | 84.30    | 76.00     | 80.12     |
> | + ConvProj                                | 84.85    | 76.80     | 80.72     |
> | + Consistency loss                        | **85.07** | **77.41** | **81.05** |
>
>
> 3. We appreciate the feedback regarding Figure 1 and have revised it to highlight our losses, components. The figure caption has also been expanded to include more comprehensive details, ensuring better understanding and clarity.
>
> 4. CasPL [1] is not directly comparable to our model due to its use of the larger CLIP-L/14 model for distilling adaptable prompts. CasPL employs a two-stage process: first, adaptable prompts are learned from the larger CLIP-L/14 model through unsupervised learning; second, a separate set of boosting prompts is trained using prior prompt-tuning methods. In contrast, our approach does not rely on distillation from larger models and instead utilizes CLIP-B/16, aligning with other comparable prompt-tuning methods. Additionally, since CasPL was released on September 26, 2024, very close to the ICLR deadline, it was not feasible to refer it in our paper.
>
> 5. Equation 5 is applied to prompted anchor view patches and zero-shot anchor view patches and does not appear to require correction. However, please let us know if we have overlooked any specific issues. Regarding Equation 9, Voronoi clustering can be applied regardless of order, as the prompts are randomly initialized, and the clustering outputs are added as a bias. These prompts are then passed through the attention mechanism, which has order invariance.
>
> [1] Cascade Prompt Learning
>
> We have updated our paper to further address your concerns:
>
> - The experiments mentioned above are included in the appendix, specifically in Tables 15, 16, 17, and 18, with detailed explanations provided in the corresponding paragraphs.
> - We revised the related works section to address [1], clearly outlining the differences between our work and the referenced paper. Additionally, we explained why comparing the two would be unfair.

---

> ### Author Response · Authors · 2024-11-29
>
> Dear Reviewer,
>
> Thank you for your detailed review of our submission. As the deadline for the rebuttal period approaches, we kindly request your feedback on our responses to ensure that we have addressed your concerns. If further clarification or additional experiments are needed, we would be happy to provide them promptly.
>
> If our responses have resolved your concerns, we would sincerely appreciate it if you could consider updating your scores accordingly.
>
> Your input is critical to the final evaluation of our work, and we look forward to hearing from you soon.

---

> ### Author Response · Authors · 2024-12-01
> **Further explanation**
>
> Dear Reviewer,
>
> Thank you for your response and for reconsidering your score. We appreciate your acknowledgment of the effort and potential of our method. We understand your concern regarding the perceived complexity of the algorithm and would like to provide a simple explanation to clarify the necessity and uniqueness of each component.
>
> **1. Why are multiple components required?**
> In few-shot learning, simultaneously regularizing vision, text, and predictions is critical but challenging (e.g., PromptSRC, CoPrompt). A simple regularization strategy would fail to account for the interdependencies between these modalities and variations across views. Integrating patch-level information into all three modalities is the only viable approach to achieve robust alignment and generalization. Without such a design, the model risks underperforming in real-world few-shot scenarios. (Experimental justification: Refer components table in above comment)
>
> **2. Why are multiple patch losses necessary?**
> A single inter-view patch loss is insufficient as it could lead to trivial solutions, such as all patches in one view aligning to a single patch in another view. The intra-view patch loss mitigates this by ensuring that patches within the same view maintain meaningful relationships, complementing the inter-view loss. This dual approach is essential to effectively inject patch-level information while preserving cross-view consistency.  (Experimental justification: Refer vision-level table in above comment)
>
> **3. Why introduce view-tailored text?**
> To regularize the text branch and ensure cross-view consistency, fine-grained patch-level information must be incorporated into text prompts. This design ensures that text prompts are tailored for each view, capturing finer image details. Regularizing these prompts across views is essential, as they stem from the same image. A design without this mechanism would fail to incorporate patch-level information coherently at the textual level.  (Experimental justification: Refer text-level table in above comment)
>
> **4. Why is prediction-level consistency necessary?**
> As demonstrated by PromptSRC, prediction-level consistency enhances cohesive alignment across modalities. However, applying this consistency solely to class token vision features leads to trivial solutions. By incorporating patch-level information into the predictions and enforcing consistency across predictions from different views, our method ensures meaningful and robust consistency, avoiding oversimplified outcomes.   (Experimental justification: Refer prediction-level table in above comment)
>
> **Why this approach is essential:**
> Our design choices are not just logical—they are necessary. Each component addresses a critical gap in existing methods and collectively forms a framework capable of solving the unique challenges of few-shot learning. Simplifying the design further would compromise performance and fail to address key issues such as cross-view consistency and sensitivity to patch-level information.
>
> **Impact and Future Directions:**
> This approach sets a new direction for the field of prompt learning by demonstrating the value of integrating patch-level information for adapting VL models. While the perceived complexity may seem significant, it is an inevitable outcome of addressing these challenges comprehensively. We believe the broader research community will refine and build upon this foundation over time, as has been the case with similar advancements in other domains.
>
> We kindly request that you reconsider your score based on this explanation. Additionally, if there are further points requiring clarification, we would be happy to provide additional details or experiments.
>
> Thank you for your time and thoughtful engagement.

---

### Official Review · Reviewer_1Nyy · 2024-11-07

**Soundness:** 3
**Presentation:** 3
**Contribution:** 3
**Rating:** 6
**Confidence:** 4

**Summary:**

This paper propose a prompt tuning method for CLIP. Unlike previous methods that use global feature consistency, the proposed method use the local feature consistency together. By incorporating the local features effectively, the proposed method achieve strong fine-tuning performance on various benchmarks

**Strengths:**

+ The proposed method is reasonable and sounds interesting.

+ The authors conducted heavy ablation studies which is helpful to understand the proposed method.

+ The proposed method achieve strong performance on various benchmark and various settings.

**Weaknesses:**

- The performance improvement over baseline seems marginal. The proposed method brings less than 0.5%p compared to baseline for most of evaluating settings. I suggest the authors to conduct repeated experiments and present the standard deviation, and show that the proposed method is statistically significant

- The proposed method has many choice of hyper-parameters and designs, which may not be appropriate for data limited learning.

**Questions:**

1, In section 3.2, the authors find the closest zero-shot patch from the anchor view using cosine similarity. However, because we know the augmentation parameters to generate the novel view, we already know the exact matching between anchor and augmented images. Why the authors find the closest patch using cosine similarity? How if we find the matching patch using the augmentation parameters?

2. In Eq (10), is g^i the class token from visual encoder? Then, what is the motivation of the sum of the global and local features in Eq 10? Can we try other operation like dot-product?

---

> ### Author Response · Authors · 2024-11-18
>
> Thank you for your valuable feedback.
>
> 1. We would like to highlight that all experiments are conducted with different seeds, consistent with previous methods. These methods report only the mean and do not include the standard deviation, making replicability and comparison of standard deviation challenging. While we acknowledge that the performance on base-to-novel generalization and domain generalization shows a slight margin of improvement, it aligns with the trend of marginal gains observed in previous methods. In contrast, our main results on the base-to-novel task demonstrate a significantly better average improvement.
> 2. While we agree that the choice of hyperparameters is complex, it enables more effective tuning for individual datasets. For instance, a dataset with significant patch variation can benefit from a higher $\lambda_p$, while larger values for $\lambda_t$ and $\lambda_l$ are better suited for datasets requiring stronger inter-view consistency at a global level.
> 3. In Section 3.2, the cropping is performed using PyTorch's `RandomResizedCrop`, which does not ensure alignment with the patch size of 16. This makes it infeasible to calculate correspondence directly based on cropping parameters. Instead, we determine correspondence by identifying the most similar patch at the feature level.
> 4. The local and global views are defined by the crop ratios of different views. Both the augmented view and the anchor view are global views by default. In Table 10, we aim to demonstrate the effect of having more than one augmented view and the impact of using a much smaller crop size. This experiment does not correspond to Eq(10) since the number of patch clusters is not varied.

---

> ### Author Response · Authors · 2024-11-21
>
> We have updated the paper to include a description of using different seeds, detailed in the "Additional Implementation Details" section of the appendix.

---

### Official Review · Reviewer_8PST · 2024-11-10

**Soundness:** 4
**Presentation:** 4
**Contribution:** 2
**Rating:** 6
**Confidence:** 4

**Summary:**

The paper introduces a new approach to prompt tuning for large vision-language models, leveraging patch-level information rather than global image semantics alone. In terms of technical design, it incoperates three key techniques:
Patch-based Consistency Loss: Ensures fine-grained regional alignment in image patches across different views.
Text Prompt Consistency: Creates view-specific text prompts to maintain cross-view coherence.
Vision Feature Regularization: Integrates patch-level details to enhance prediction quality.
The new method shows gains on three setups: base to novel, cross datasets, and domain generalization.

**Strengths:**

1. The combination of techniques showed gains on all nearly all eval metrics in all the three setups.
2. The authors conducted many ablation studies regarding different components and design choices of the model. They are very informative.

**Weaknesses:**

Overall, the final solution seems to be too complicated. There are quite a few components, losses, and tricks. There was not always a good explanation of why a certain component/trick would help. Some ablation experiments are still missing. The number of trained parameters is much higher than PromptSRC and DePT. The training cost is also much higher especially given that an augmented view is needed on top of the original view. The eval cost is also much higher since the text embedding needs to be conditioned on every image. More specifically:

1. How much can the inter-view patch loss help on top of the intra-view patch loss? This design greatly increases the training cost, so it would be great if the effect of this design is shown.
2. Why is the convolution projection layer of patch features needed? Table 9 showed that it further improved some eval numbers. But what is the rationale of those gains? Are the gains worthy the additional complexity of the system? Especially given that it is not just a MLP, but a 3x3 convolutional block added on top of a visual transformer. Why not one additional transformer block?
3. The authors chosed to condition the text embedding on image patch features. This greatly increased the eval/inference cost. Let's say there are N images and M classes. Without that conditioning, we only need to compute N image embeddings and M text embeddings. But since the text embedding needs to be conditioned on the image that is in question, so now we need to compute N image embeddings and MxN text embeddings. Moreover, this conditioning design made the addition of the clustering algorim necessary as well.
4. The authors chose to add the average of the patch features to the image-level features to get the final image embedding. Are any of the embeddings normalized? Could the authors elaborate on their design regarding which features are normalized and which are not, and explain why this deisgn choice?

**Questions:**

The original motivation of the paper is just to add patch-level information into prompt tuning. But the final design is much more complex than what is needed to involve patch-level information in prompt tuning. Is it that a very simple design would not show gain? Then does it suggest that the original motivation is not very convincing? The final solution showed gains, but was it just because it added patch-level information, or was it because all the increased system complexity and added trainable parameters?

---

> ### Author Response · Authors · 2024-11-18
>
> Thank you for your valuable feedback.
>
> While we believe that the paper has multiple components, carefully ablation studies have been performed to justify our choices. Evenmore ablations for simple choices could have performed, though we believe it would take exponential time for doing it.
>
> 1. Inter-view and intra-view consistency alone do not suffice for achieving optimal performance; their combination proves most effective as shown below. The results without inter-view and intra-view consistency are derived using only a single anchor view. Intra-view consistency adds an extra layer of regularization by aligning with zero-shot CLIP patches. While CLIP is designed to produce consistent global representations across different image crops, patch-level information can vary significantly. To address this, our inter-view patch loss enhances generalizability by preserving the relationships between patches across two views, ensuring closer alignment with zero-shot CLIP outputs. With the intra-view patch loss in place, the inter-view patch loss introduces an additional layer of non-triviality by enforcing consistency across views. Conversely, the presence of intra-view patch loss prevents solutions from straying too far from the foundational CLIP zero-shot patch knowledge, ensuring the inter-view patch loss does not lead to unintended deviations. Additionally, this inter-view patch knowledge is successfully extended to enforce consistency in the text branch and to refine predictions, validating the effectiveness of our approach.
>
> | Intra  | Inter  | Base   | Novel  | HM     |
> |--------|--------|--------|--------|--------|
> | ✗      | ✗      | 84.27  | 75.58  | 79.68  |
> | ✓      | ✗      | 84.53  | 75.71  | 80.03  |
> | ✗      | ✓      | 84.31  | 75.89  | 80.01  |
> | ✓      | ✓      | 85.07  | 77.41  | 81.05  |
>
>
> 2. We adopt convolutional projection as it provides superior local context compared to transformers. While transformers are effective for capturing global context, we aim to aggregate local contextual information to model finer concepts. Additionally, attention mechanisms in transformers are more memory-intensive than convolutions due to self-attention, even though the runtime remains similar. The gains shown in Table 9 highlight the advantages of this approach over MLP adapters.
> 3. As highlighted in point 1, the patch-level relationships across views are crucial for enhancing generalizability. Building on this insight, the only viable way to extend this principle to the text modality is by conditioning text with patch-level information. While clustering is indispensable for achieving the fine-grained conditioning required, it neither impacts training efficiency nor imposes additional memory overhead. This makes it a highly effective strategy, yielding significant gains in performance and generalizability.
> 4. We have applied L2 normalization to both our patch and class embeddings, a technique proven effective in prior works. This normalization enhances the stability and alignment of embedding..
> 5. Our work is designed to incorporate patch-level information while also leveraging patch relationships across views as a cue for regularizing text, vision features, and predictions. Simply focusing on integrating patches in isolation, whether in vision, text, or predictions, is insufficient, as demonstrated in Table 4. Additional results shown below compare a model with similar parameters to ours but trained without any of our proposed components. These findings highlight that simply increasing the number of parameters does not improve performance, emphasizing the importance of our approach.
>
> | Components       | Base   | Novel  | HM     |
> |------------------|--------|--------|--------|
> | No Components    | 84.55  | 75.43  | 79.89  |
> | Our Components   | **85.07** | **77.41** | **81.05** |

---

> ### Author Response · Authors · 2024-11-21
>
> We have updated our paper to address your concerns in greater detail:
>
> - We included an additional ablation study in the appendix (Table 15) to justify the effectiveness of our proposed components. In this study, we trained models with identical parameters, both with and without our components. The results demonstrate a significant performance improvement when our components are included.

---

### Author Response · Authors · 2024-11-21

We sincerely thank the review committee for their time and for providing constructive feedback. We have carefully addressed all the concerns raised by the reviewers under the individual response section. We have also updated the main PDF. Following, we provide a summary of our responses:

- **Regarding complexity**: To alleviate concerns regarding complexity raised by **Reviewer P4bj** and **Reviewer 8PST**, we conducted an additional ablation study, presented in Appendix Table 15, to justify the effectiveness of our components. Models with identical parameters were trained both with and without these components. The results demonstrate a significant performance improvement when our components are included. Additional Tables for component level choices in vision, text and predictions are provided in Table 16, 17, 18.

- **Regarding PromptSRC and CoPrompt**: Addressing concerns raised by **Reviewer 63du** and **Reviewer u15T**, we provided a clearer explanation of the differences with PromptSRC and CoPrompt in the third paragraph of the introduction. The updated section emphasizes that our method addresses consistency at a finer level and incorporates inter-view consistency across vision, text, and predictions.

- **Figure 1 Improvement**: In response to comments from **Reviewer P4bj** and **Reviewer u15T**, we revised Figure 1 to better highlight our losses and components. The figure caption has been expanded with more comprehensive details to ensure improved clarity and understanding.

- **Regarding CasPL**: To address **Reviewer P4bj**’s comment on CasPL[1], we added a detailed explanation in the Related Works section, clarifying that comparisons are unfair due to CasPL’s reliance on the larger CLIP-L/14 model.

- **Regarding [2]**: In response to **Reviewer u15T**’s comment on [2], we updated the Related Works section to note that it was independently developed. While it uses patch clusters for text conditioning, it does not address patch consistency or inter-view consistency across vision, text, and predictions as our method does.

- **Regarding Equations**: To address **Reviewer u15T**’s concerns about equations being similar to prior works, we provided detailed explanations in the relevant methodology paragraphs, outlining the exact differences.

- **Regarding DePT**: Responding to **Reviewer 63du**’s comments on DePT, we added more details about its integration in the "Additional Implementation Details" section of the appendix. Additionally, to address concerns from **Reviewer 1Nyy**, we clarified that different seeds were used in our experiments to ensure robust results.

[1] Cascade Prompt Learning
[2] Mutual Prompt Leaning for Vision Language Models

---

### Meta-Review · Area_Chair_G8SM · 2024-12-18

**Metareview:**

This paper proposes an approach to prompt tuning that leverages local semantics by incorporating patch-level information. This approach is conducted by aggregating patch-level information across vision, text, and predictions through three consistency mechanisms. The results show promising performance compared to baseline methods, and the presentation quality is solid.

However, the proposed method is of high complexity, though the reason for adding each component can be justified. Directing incorporating these losses (mechanisms) into training is intuitive but not novel enough. It would be great if the method could be simplified and thoroughly investigated to meet the bar of ICLR. Therefore, I suggest rejection of the paper.

**Additional Comments On Reviewer Discussion:**

The main discussion point of the paper was centered around the complexity of the method:
- Reviewer 8PST is concerned about the additional cost because of the intra-view loss and the patch embedding for text conditioning.
- Reviewer P4bj is concerned about the difficulty of analysis and implementation.

Moreover, reviewer 1Nyy is concerned about the marginal quality gain of the method.

Though the authors have tried their best to address the above questions with detailed experiments and explanations, the complexity of the method and the resulting performance are still the weaknesses of this work.

---

### Decision · Program_Chairs · 2025-01-22

Reject